# Progression of chronic pain and associated health-related quality of life and healthcare resource use over 5 years after total knee replacement: evidence from a cohort study

Sophie Cole [ID],[1] Spyros Kolovos,[1] Anushka Soni [ID],[1] Antonella Delmestri,[1] Maria T Sanchez-Santos [ID],[1] Andrew Judge,[1,2] Nigel K Arden,[1] Andrew David Beswick [ID],[2,3] Vikki Wylde,[2,3] Rachael Gooberman-Hill [ID],[2,3] Rafael Pinedo-Villanueva [ID] [1]

[1]Nuffield Department of Orthopaedics, Rheumatology and Musculoskeletal Sciences, University of Oxford, Oxford, UK
[2]National Institute for Health Research Bristol Biomedical Research Centre, University of Bristol, Bristol, UK
[3]Musculoskeletal Research Unit, Translational Health Sciences, Bristol Medical School, University of Bristol, Bristol, UK

**Correspondence to**
Dr Rafael Pinedo-Villanueva;
rafael.pinedo@ndorms.ox.ac.uk

## ABSTRACT

**Objective** As part of the STAR Programme, a comprehensive study exploring long-term pain after surgery, we investigated how pain and function, health-related quality of life (HRQL), and healthcare resource use evolved over 5 years after total knee replacement (TKR) for those with and without chronic pain 1 year after their primary surgery.

**Methods** We used data from the Clinical Outcomes in Arthroplasty Study prospective cohort study, which followed patients undergoing TKR from two English hospitals for 5 years. Chronic pain was defined using the Oxford Knee Score Pain Subscale (OKS-PS) where participants reporting a score of 14 or lower were classified as having chronic pain 1-year postsurgery. Pain and function were measured with the OKS, HRQL using the EuroQoL-5 Dimension, resource use from yearly questionnaires, and costs estimated from a healthcare system perspective. We analysed the changes in OKS-PS, HRQL and resource use over a 5-year follow-up period. Multiple imputation accounted for missing data.

**Results** Chronic pain was reported in 70/552 operated knees (12.7%) 1 year after surgery. The chronic pain group had worse pain, function and HRQL presurgery and postsurgery than the non-chronic pain group. Those without chronic pain markedly improved right after surgery, then plateaued. Those with chronic pain improved slowly but steadily. Participants with chronic pain reported greater healthcare resource use and costs than those without, especially 1 year after surgery, and mostly from hospital readmissions. 64.7% of those in chronic pain recovered during the following 4 years, while 30.9% fluctuated in and out of chronic pain.

**Conclusion** Although TKR is often highly beneficial, some patients experienced chronic pain postsurgery. Although many fluctuated in their pain levels and most recovered over time, identifying people most likely to have chronic pain and supporting their recovery would benefit patients and healthcare systems.

## Strengths and limitations of this study

► A strength of the study is that participant-level data were collected over 5 years after total knee replacement enabling a detailed analysis of changes over time.
► The study cohort offered data on patient-reported outcome measures, allowing for a classification of participants according to their chronic pain status following surgery, and linked healthcare resource use to include in the analysis.
► This study followed pain trajectories of participants with postsurgical chronic pain which has not previously been explored.
► An important limitation was that many follow-up questionnaires were not returned, which generated an important level of missing data, addressed using standard methods of multiple imputation.
► The longitudinal study questionnaires did not ask participants about informal care, productivity losses or their use of privately funded healthcare other than physiotherapists, which may play an important role in associated costs.

## INTRODUCTION

Total knee replacement (TKR) provides pain relief and increases function for many people with advanced-stage knee osteoarthritis, which improves health-related quality of life (HRQL). However, a distinct, important group of people report long-term (chronic) pain after their TKR.[1–5] Chronic postsurgical pain is defined as pain that persists at least 3 months after surgery, beyond the healing process.[6] There is currently limited understanding of how chronic pain (CP) after a TKR changes over time, shapes people's lives, and affects HRQL and healthcare service use.

Although pain trajectories after TKR have previously been explored,[7] those who report postsurgical CP have not been followed specifically. It is not yet known what proportion of individuals who experience CP in the first year after TKR find that their pain dissipates over the following few years, remain in CP, or find that their pain fluctuates. As CP is a complex construct, it would be useful to compare people who transition between these groups to understand whether they are more similar to the group they left or the group they joined. Better understanding of CP trajectories would help discern how CP evolves, giving patients clearer information about how likely they are to recover from their CP, and helping clinicians gain greater insight into the condition, hopefully contributing to finding ways to treat patients more effectively.

Assessing clinical and cost-effectiveness is a critical step in informing decision-making about the wider implementation of new interventions. For cost-effectiveness studies of potential new interventions for CP following surgery to be carried out, our understanding of the progression of CP must be accompanied by an assessment of HRQL and costs. HRQL is affected by a number of factors including the severity of pain and its consequences and is expected to vary as people's pain improves, worsens or fluctuates. Changes in pain are also likely to impact healthcare use, such as follow-up hospital visits, general practitioner (GP) appointments, or prescriptions of analgesia. Healthcare resource use and costs after a TKR have been previously described[8 9] but, to our knowledge, these have not been examined specifically for those in CP, hence leaving an important research gap which needs to be closed to enable future cost-effectiveness assessments of interventions addressing CP after TKR. This study aimed to describe how pain and function, HRQL, and healthcare resource use evolved over the first 5 years after TKR, for patients with and without CP 1 year after surgery, using data collected in a published prospective cohort study.[9] A secondary objective was to map how the CP status of those with CP changed and assess how changes in HRQL and healthcare costs for those recovering from CP after the first year compared with those who did not recover or never had CP.

## PATIENTS AND METHODS
### Clinical Outcomes in Arthroplasty Study
The Clinical Outcomes in Arthroplasty Study (COASt), a prospective cohort study, tested outcome prediction models' performance[9] by collecting preoperative and postoperative outcome data from patients undergoing hip and knee replacements with yearly follow-up. It was extended to 5 years to capture long-term pain and function, HRQL, and healthcare resource use.

COASt recruited patients receiving a joint replacement from the Nuffield Orthopaedic Centre in Oxford from 2010 and the Southampton General Hospital from 2011 (both in England). Patients were invited to participate after they were placed on the waiting list for a primary or revision knee replacement regardless of age, gender, body mass index (BMI) or underlying cause for the surgery. Baseline information was collected in preoperative questionnaires that covered participant demographics such as the Index of Multiple Deprivation (a measure of relative deprivation at a small local area level in England) and patient-reported outcome measures (PROMs): HRQL (EuroQuol-5 Dimension-3 Levels, EQ-5D-3L) and pain and function (Oxford Knee Score, OKS). COASt participants were followed up with yearly postal questionnaires for 5 years postsurgery.

We included COASt participants who underwent a primary TKR, returned their questionnaires 1 year after surgery and completed the OKS Pain Subscale (OKS-PS) in that questionnaire for the resource use analysis. We excluded bilateral surgeries to be able to track levels of pain of a single operated knee. We did not have information about which patients had a revision surgery, which is highly unlikely during the first years after primary, or whether they had a contralateral operation. For the pain, function and HRQL analysis we also included participants who did not complete their OKS-PS at 1 year, in which cases missing data were imputed. Patients listed for hip or knee replacement surgeries in the two participating centres (Southampton and Oxford) were potentially eligible for inclusion. Once identified, potential participants were sent a recruitment pack, consisting of a patient information sheet, a sample consent form and a recruitment letter. They were later contacted by telephone by a member of the COASt team around 2 weeks after the pack was sent to discuss study details and answer any question they might have had. The COASt team member then took verbal consent if patients were satisfied and willing to participate in the study.[9]

### Patient and public involvement
Patient and public involvement (PPI) members were actively involved in the design and recruitment of the STAR programme. PPI members participated in over 20 meetings of the STAR Patient Forum to discuss and improve the participant experience. They gave feedback and recommended changes for both recruitment and study materials for the STAR trial, which was further informed by the results of this study. Findings were disseminated to study participants via regular programme bulletins and through STAR Patient Forum meetings.

### Pain and function
Pain and functional ability were measured with the OKS, a patient-reported questionnaire consisting of 12 items assessing a patient's perspective of their joint pain and function after a TKR.[10] Each item has five possible responses, with higher values indicating better outcomes, and contributes 0–4 to the total score, which ranges 0–48.

The OKS-PS was used to identify those in CP, following work by some of our co-authors. The OKS-PS includes the

7 OKS questions relating to pain and is summarised by a score of 0–28.[11] Participants reporting an OKS-PS of 14 or lower 1 year after surgery were classified as having CP, while those scoring above 14 were classified as non-CP.[4] We considered CP 1-year postoperatively to give patients sufficient time to recover from the operation and ensure that any pain reported was not due to the operation itself. While those classified as having no-CP may still experience some pain, they have been found to report much higher quality of life and greater satisfaction with the result of the operation.[4]

### Health-related quality of life

HRQL was measured using the EQ-5D-3L questionnaire, which examines five dimensions of health: mobility, self-care, usual activities, pain/discomfort and anxiety/depression. This instrument was completed at baseline (just before surgery) and annually after the TKR for 5 years. A health utility estimate, anchored at 0 representing death and with 1 representing full health, was calculated by applying a social preference tariff collected from a sample of the UK general population.[12]

### Healthcare resource use

Healthcare resource use was measured using participants' answers to questions about their visits to GPs, nurses, physiotherapists, alternative practitioners and admissions to hospitals, because of their operated knee. These questions were included in the baseline and all follow-up questionnaires and covered the preceding 12 months.

The value of healthcare resources was estimated from the National Health Service (NHS) perspective by applying mean unit costs sourced from the Personal Social Services Research Unit[13] and National Cost Collection.[14] To calculate the cost of readmittance to hospital, we used a binary variable and information from a free text section in the questionnaire explaining the readmittance. The submitted free text was used to identify participants who were treated for a knee infection or underwent a knee procedure. Unit costs were applied according to this categorisation and costs estimated by taking a weighted average across a number of reported knee procedures and treatments for knee infections.[14]

### Missing data

Lost to follow-up during the 5 years led to increasing amounts of missing data. Missing data and attrition are an important concern for longitudinal studies. It has been argued that the imputation of missing covariates data in medical research is always better than the complete case.[15] Excluding subjects with missing values leads to a reduction in the sample study size and may diminish the predictive power of the working statistical model.

Therefore, multiple imputation assuming data missing at random was implemented to reduce the potential biases arising from missing data. Fifty imputed

datasets were generated, with missing OKS-PS and EQ-5D values replaced by imputed ones.[16] We used the imputation by chain equation and applied the predictive mean matching two-level imputation method to account for each participant completing the follow-up questionnaires multiple times. To ensure that imputation did not introduce bias into the results, we compared the outcome distribution in the observed and imputed datasets.[17] We also analysed the participant demographics for the observed and missing data at 5-year follow-up, how many years the participants had missing OKS-PS or EQ-5D values, and investigated our assumption of the missing data being missing at random with logistic regression. We censored participants who died within the 5 years of follow-up (n=30) so that we did not impute data beyond their death. Summary statistics for the imputed dataset's outcomes were combined with Rubin's rule.[18]

### Analysis

We characterised HRQL and pain outcomes, resource use, and costs by CP group (CP and non-CP, based on 1-year postoperative outcomes) over 5 years by reporting the mean and SD of the imputed OKS, OKS-PS, health utility, number of healthcare visits and healthcare costs from presurgery to 5 years postsurgery. In addition, for OKS-PS and health utility we reported 95% confidence intervals. We considered presurgery as well as postsurgery to investigate whether those with CP postsurgery report distinct differences even prior to surgery.

To investigate the trajectory of CP, we tracked CP-group participants' movement in and out of CP and report the percentage that remained in CP for the entire period, recovered from CP at any point, and fluctuated between CP and non-CP over the 5 years. A participant's CP group status at each time point was determined by their observed OKS-PS, if available, or the mean across the 50 imputed scores, if missing. Transitions in and out of CP are illustrated in a Sankey diagram.

To investigate whether leaving the CP group after year 1 led to changes in HRQL or costs, we compared observed health utility and healthcare costs of CP-group participants who recovered from CP by year 2 after surgery and those who did not.

All analyses were conducted in R V.4.0.3[19] using multiple packages for data cleaning and statistical analysis,[20–27] performing multiple imputation[28–31] and producing figures.[32–34] We followed the Strengthening the Reporting of Observational Studies in Epidemiology statement as a reporting guideline for this study.[35]

### RESULTS

Of the 1025 knee procedures enrolled in COASt, all 580 corresponding to a TKR returned their year-1 follow-up questionnaire. We excluded 28 who did not report their OKS-PS in that questionnaire, giving a final sample of 552. Seventy (of 552, 12.7%) were

**Table 1** Study participant demographics

| Variable | Total at Y1 (n=552) | | | CP at Y1 (n=70) | | | Non-CP at Y1 (n=482) | | | |
|---|---|---|---|---|---|---|---|---|---|---|
| **Age at total knee replacement** | **N** | **Mean** | **Range** | **N** | **Mean** | **Range** | **N** | **Mean** | **Range** | **P value** |
| Total | 552 | 70 | (38–90) | 70 | 70 | (42–88) | 482 | 70 | (38–90) | 0.914 |
| Below 61 | 62 | 54 | (38–60) | 7 | 54 | (42–58) | 55 | 54 | (38–60) | |
| 61–70 | 210 | 66 | (61–70) | 28 | 66 | (61–70) | 182 | 66 | (61–70) | |
| 70–80 | 227 | 75 | (71–80) | 27 | 75 | (71–79) | 200 | 75 | (71–80) | |
| Above 80 | 53 | 84 | (81–90) | 8 | 84 | (81–88) | 45 | 84 | (81–90) | |
| Gender | N | % | | N | % | | N | % | | |
| Female | 308 | 55.8 | | 48 | 68.6 | | 260 | 53.9 | | 0.021 |
| Male | 244 | 44.2 | | 22 | 31.4 | | 222 | 46.1 | | |
| IMD 2010 decile | N | % | | N | % | | N | % | | |
| 1 (least deprived) | 122 | 22.1 | | 15 | 21.4 | | 107 | 22.2 | | 0.187 |
| 2–5 | 294 | 53.3 | | 33 | 47.1 | | 261 | 54.2 | | |
| 6–9 | 130 | 23.6 | | 20 | 28.5 | | 110 | 22.8 | | |
| 10 (most deprived) | 5 | 0.9 | | 2 | 2.9 | | 3 | 0.6 | | |
| Missing (n, (%)) | 1 | (0.2) | | 0 | 0 | | 1 | (0.2) | | |
| BMI | N | Mean | SD | N | Mean | SD | N | Mean | SD | |
| Total | 548 | 30.7 | 5.5 | 70 | 32.4 | 5.9 | 478 | 30.4 | 5.4 | 0.010 |
| Below 25 | 76 | 23.1 | 1.6 | 6 | 22.2 | 2.5 | 70 | 23.1 | 1.5 | |
| 25–29.9 | 199 | 27.6 | 1.4 | 18 | 27.9 | 1.4 | 181 | 27.6 | 1.4 | |
| 30–34.9 | 148 | 32.2 | 1.4 | 21 | 32.0 | 1.5 | 127 | 32.2 | 1.4 | |
| Above 34.9 | 125 | 38.5 | 3.4 | 25 | 38.4 | 3.9 | 100 | 38.5 | 3.3 | |
| Missing (n, (%)) | 4 | (0.7) | | 0 | (0.00) | | 4 | (0.8) | | |

Differences in continuous and categorical variables between the CP and non-CP groups were tested with a t-test and $\chi^2$ test, respectively.
BMI, body mass index; CP, chronic pain; IMD, Index of Multiple Deprivation.

classified as in CP 1 year after surgery using the OKS-PS. We refer to those classified as in CP 1 year after TKR as the CP group, and those not in CP 1 year after TKR as the non-CP group.

Table 1 shows that the CP and non-CP groups had similar demographics. Both had a mean age of 70 years at surgery and similar mean BMIs, although higher in the CP group. A greater proportion of the CP group were female (69%) than the non-CP group (54%). Both groups comprised a higher proportion of people living in less deprived areas than the national distribution. The CP group reported worse preoperative pain, function and HRQL scores than the non-CP group (table 2). In addition, similar statistically significant differences were observed with linear regressions controlling for both gender and BMI. The amount of missing data due to lost to follow-up is reported in tables 1 and 2.

### Characterisation of HRQL and pain outcomes, resource use and costs

Observed and imputed data showed similar mean health utility and OKS-PS scores (online supplemental table A.1 and online supplemental figures A.1 and A.2). Observed and missing data at 5-year follow-up reported similar preoperative demographics and mean health utility and OKS-PS scores (online supplemental table A.2). 77.9% of participants had between 0 and 2 missing years of OKS-PS across the 5 years of follow-up (online supplemental table A.3). Those in CP reported a higher number of years of missing items for both OKS-PS and EQ-5D (online supplemental figures A.3 and A.4). Logistic regression showed that dimensions of EQ-5D and OKS-PS measured preoperatively had a statistically significant association with the missing OKS-PS across the 5 years of follow-up (online supplemental table A.4).

The CP and non-CP groups had different mean health utility estimate scores before surgery (figure 1) although their confidence intervals overlapped (online supplemental table A.5). Participants in CP 1 year after TKR had a preoperative health utility estimate of 0.307, whereas those who would not be in CP had a preoperative score of 0.485. The two groups had noticeably different changes in health utility over the next 5 years. On average, the non-CP group improved significantly during the first year after surgery, reaching a score of 0.787 (compared with a preoperative score of 0.485), and stayed at a similarly high level until follow-up ended (0.751 after 5 years). The

**Table 2** Study participants health outcomes

| Variable | Total at Y1 (n=552) | | | CP at Y1 (n=70) | | | Non-CP at Y1 (n=482) | | | P value |
|---|---|---|---|---|---|---|---|---|---|---|
| | N | Mean | SD | N | Mean | SD | N | Mean | SD | |
| **OKS** | | | | | | | | | | |
| Baseline | 490 | 19.4 | 7.7 | 62 | 14.1 | 6.8 | 428 | 20.2 | 7.5 | <0.001 |
| Missing (n, (%)) | 62 | (11.2) | | 8 | (11.4) | | 54 | (11.2) | | |
| Year 1 | 539 | 36.2 | 10.0 | 67 | 16.9 | 6.0 | 472 | 38.9 | 6.9 | <0.001 |
| Missing (n, (%)) | 13 | (2.4) | | 3 | (4.3) | | 10 | (2.1) | | |
| Year 2 | 435 | 37.8 | 9.5 | 44 | 22.1 | 8.7 | 391 | 39.6 | 7.8 | <0.001 |
| Missing (n, (%)) | 117 | (21.2) | | 26 | (37.1) | | 91 | (18.9) | | |
| Year 3 | 386 | 38.2 | 9.0 | 41 | 24.4 | 9.3 | 345 | 39.8 | 7.4 | <0.001 |
| Missing (n, (%)) | 166 | (30.1) | | 29 | (41.4) | | 137 | (28.4) | | |
| Year 4 | 320 | 38.6 | 8.9 | 30 | 26.0 | 10.4 | 290 | 39.9 | 7.6 | <0.001 |
| Missing (n, (%)) | 232 | (42.0) | | 40 | (57.1) | | 192 | (39.8) | | |
| Year 5 | 274 | 38.3 | 9.2 | 20 | 25.3 | 9.6 | 254 | 39.4 | 8.3 | <0.001 |
| Missing (n, (%)) | 278 | (50.4) | | 50 | (71.4) | | 228 | (47.3) | | |
| **OKS-PS** | | | | | | | | | | |
| Baseline | 493 | 10.2 | 4.7 | 62 | 7.1 | 4.0 | 431 | 10.6 | 4.6 | <0.001 |
| Missing (n, (%)) | 59 | (10.7) | | 8 | (11.4) | | 51 | (10.6) | | |
| Year 1 | 552 | 22 | 6.1 | 70 | 9.7 | 3.6 | 482 | 23.8 | 4.0 | <0.001 |
| Missing (n, (%)) | 0 | (0.0) | | 0 | (0.0) | | 0 | (0.0) | | |
| Year 2 | 448 | 23.1 | 5.7 | 44 | 13.4 | 5.8 | 404 | 24.2 | 4.6 | <0.001 |
| Missing (n, (%)) | 104 | (18.8) | | 26 | (37.1) | | 78 | (16.2) | | |
| Year 3 | 402 | 23.3 | 5.5 | 41 | 14.8 | 6.1 | 361 | 24.3 | 4.5 | <0.001 |
| Missing (n, (%)) | 150 | (27.2) | | 29 | (41.4) | | 121 | (25.1) | | |
| Year 4 | 341 | 23.7 | 5.4 | 31 | 15.9 | 6.6 | 310 | 24.5 | 4.5 | <0.001 |
| Missing (n, (%)) | 211 | (38.2) | | 39 | (55.7) | | 172 | (35.7) | | |
| Year 5 | 286 | 23.5 | 5.5 | 21 | 15.7 | 5.9 | 265 | 24.1 | 5.0 | <0.001 |
| Missing (n, (%)) | 266 | (48.2) | | 49 | (70.0) | | 217 | (45.0) | | |
| **Health utility estimate** | | | | | | | | | | |
| Baseline | 494 | 0.452 | 0.295 | 64 | 0.2732 | 0.315 | 430 | 0.4789 | 0.283 | <0.001 |
| Missing (n, (%)) | 58 | (10.5) | | 6 | (8.6) | | 52 | (10.8) | | |
| Year 1 | 538 | 0.74 | 0.254 | 67 | 0.3857 | 0.310 | 471 | 0.7909 | 0.199 | <0.001 |
| Missing (n, (%)) | 14 | (2.5) | | 3 | (4.3) | | 11 | (2.3) | | |
| Year 2 | 449 | 0.766 | 0.269 | 47 | 0.4422 | 0.321 | 402 | 0.804 | 0.235 | <0.001 |
| Missing (n, (%)) | 103 | (18.7) | | 23 | (32.9) | | 80 | (16.6) | | |
| Year 3 | 398 | 0.76 | 0.255 | 41 | 0.4209 | 0.314 | 357 | 0.7987 | 0.216 | <0.001 |
| Missing (n, (%)) | 154 | (27.9) | | 29 | (41.4) | | 125 | (25.9) | | |
| Year 4 | 336 | 0.761 | 0.265 | 31 | 0.4953 | 0.298 | 305 | 0.7877 | 0.247 | <0.001 |
| Missing (n, (%)) | 216 | (39.1) | | 39 | (55.7) | | 177 | (36.7) | | |
| Year 5 | 289 | 0.756 | 0.270 | 23 | 0.5398 | 0.326 | 266 | 0.7748 | 0.257 | 0.003 |
| Missing (n, (%)) | 263 | (47.6) | | 47 | (67.1) | | 216 | (44.8) | | |

Differences in continuous variables between the CP and non-CP groups were tested with a t-test.
CP, chronic pain; OKS, Oxford Knee Score; OKS-PS, OKS Pain Subscale.

CP group slowly improved their average health utility (except between years 2 and 3), going from 0.399 1 year after surgery to 0.656 5 years after surgery (figure 1 and online supplemental table A.5).

There were similar patterns in pain progression, measured with the OKS-PS (online supplemental figure A.5). The non-CP group started with a higher average preoperative score (11.3), improved significantly during

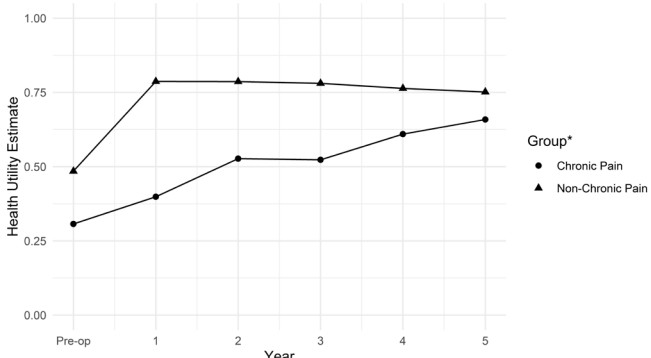

**Figure 1** Progression of health utility by chronic pain status 1 year after total knee replacement surgery, defined using a threshold score in the Oxford knee score pain subscale. *Groups were defined based on partcipants' reported OKS Pain Subscale one year after primery surgery, with those scoring 14 or less classified as having chronic pain, and those reporting scores greater than 14 as having no chronic pain.

the first year after surgery (23.9), and then stabilised (23.8 after 5 years). The CP group started with a lower average preoperative score (8.0) and slowly but steadily improved, from 9.7 after 1 year to 20.8 after 5 years (online supplemental table A.6).

Online supplemental tables A.7a and A.8b show participants' use of healthcare resources. In the year before surgery, the groups made comparable numbers of visits to an NHS GP, with the CP group reporting an average 3.8 visits and the non-CP group an average 3.4. However, their use patterns differed after surgery. One year after surgery, mean yearly visits were 2.7 for the CP group and 0.6 for the non-CP group. The mean number of NHS GP visits then fell steadily for both groups to an average of 0.3 visits for the CP group and 0.1 for the non-CP group 5 years after surgery.

The groups' average annual number of visits to physiotherapists, hospital doctors, nurses and alternative practitioners differed by healthcare specialist, but showed similar patterns. The CP and non-CP groups made similar numbers of visits before surgery; numbers surged for both groups during the 12 months after surgery, especially for the CP group. Visits then progressively decreased over the rest of the follow-up period for both groups. Although most physiotherapy visits were to NHS practitioners, during the first year postsurgery the CP group reported an average of four visits to NHS physiotherapists and three visits to private physiotherapists, whereas the non-CP group respectively made 2.2 and 0.6 visits. The groups' number of visits to Accident and Emergency, readmissions to the same hospital, and admission to any other hospital all showed similar patterns over time as the visits to non-GP healthcare professionals.

Participants in the two groups had similar mean healthcare costs in the year before surgery: £430 (SD=611.41) for the CP group and £322 (SD=337.22) for the non-CP group. Figure 2 shows the change in mean yearly costs by category over time. Consistent with the changes in resource use, mean costs increased to £1799 (SD=2981.69) for those in the

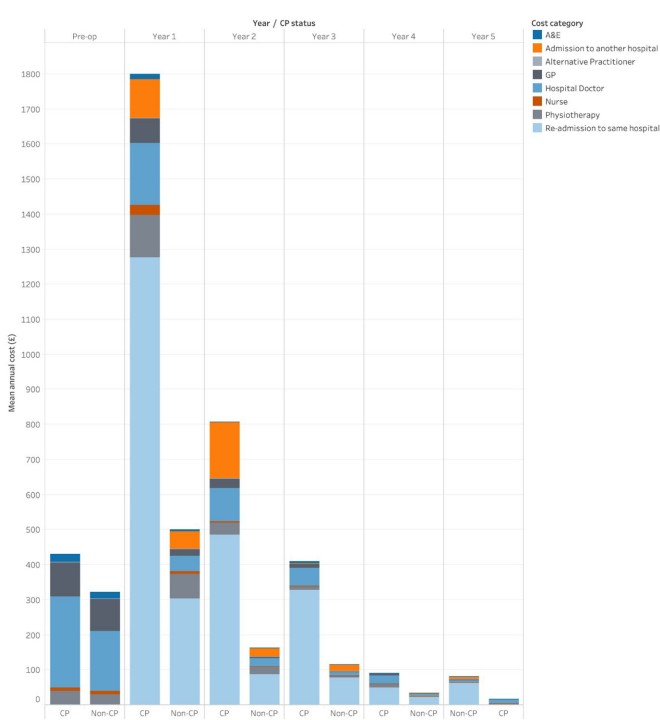

**Figure 2** Yearly healthcare costs by chronic pain (CP) status 1 year after total knee replacement surgery, defined using a threshold score in the Oxford knee score pain subscale. A&E, Accident and Emergency; GP, general practitioner.

CP group 1 year after TKR and £501 (SD=1511.82) for those in the non-CP group. Readmission to hospital was responsible for 75% of the differences between the groups (£973 of £1298) during this first year after surgery. After the first year, mean healthcare costs fell steadily for both cohorts. Online supplemental tables A.9a and A.9b show yearly healthcare costs by CP status from preoperation to 5 years after TKR.

### Progression of CP status
Figure 3 shows the change in CP status over the 5 years after TKR, based on yearly OKS-PS scores. Only 4.4% of those who reported experiencing CP after 1 year remained in CP throughout the 5 years. Almost a third of participants (30.9%) fluctuated in and out of CP. Most (64.7%) of those in CP 1-year postsurgery left CP within the next 4 years and did not experience CP again. Most of those who recovered from CP did so during the second year after surgery (26/44), with fewer people leaving CP each year thereafter except for the final year.

### Changes between year 1 and year 2
Mean health utility remained stable for those who remained in the same CP group between years 1 and 2, while those who moved into CP saw their health utility drop and those who recovered from it reported a clear improvement (table 3).

The groups' healthcare costs followed similar patterns to their health utility scores (online supplemental tables A.10 and A.11). Participants who remained in CP had the highest costs in both the first (£1600) and second (£1450) years postsurgery. Those who recovered from CP after the first year

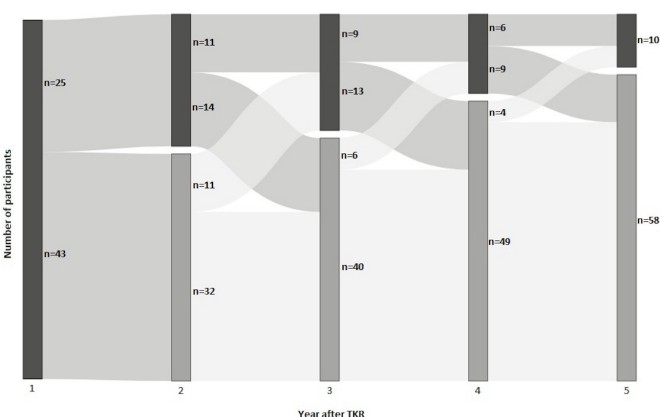

**Figure 3** Progression of chronic pain status over 5 years after total knee replacement (TKR), showing participants in chronic pain (dark grey) and not in chronic pain (light grey). Chronic pain status set using a threshold value for the mean Oxford knee score pain subscale value across 50 imputations for each participant at each year.

reported a drop in costs from £1500 in the first year to £1000 in the second year. Those who developed CP in the second year saw their costs drop from £1300 to £1200. Those who remained free of CP had the lowest costs in both the first (£481) and second (£133) years.

## DISCUSSION

Our analysis of data from COASt found that, on average, TKR led to large HRQL improvements, with health utility scores improving from 0.447 before surgery to 0.742, 1 year after surgery. This improvement is consistent with what has been reported in previous studies,[36] including average health utility gains of 0.334 over 6 months in 2018/2019 reported by the national English NHS PROMs.[37] However, these results were for all participants and hide the experience of those in CP for months after TKR. We found that 12.7% of participants reported CP 1-year after surgery, which is consistent with previous studies that used different mechanisms to ascertain CP.[1–3 5]

Our characterisation of pain and function, HRQL, and healthcare resource use and costs by participants with and without CP 1 year after surgery suggests that these groups are distinct. However, CIs are wide and overlap. Study participants in the non-CP group reported rapid, significant improvements in OKS-PS, OKS and EQ-5D over the 5 years after surgery, whereas those with CP improved much less and much more slowly. It's not surprising that the pain, composite pain and function, and HRQL measures changed in similar ways, given their close association.[38] A key distinction between the groups was their preoperative health status. Those in the CP group had lower preoperative scores, indicating worse health, than those without CP for pain (mean OKS-PS=8 and 11, respectively) and HRQL (mean health utility=0.307 and 0.485, respectively). Those in the CP group accessed more community healthcare and hospital services than those in the non-CP group at all time points, most notably during the first year after surgery

**Table 3** Mean health outcomes for fluctuating and stable groups over chronic pain (CP) status between 1 and 2 years after total knee replacement

| | n | Mean | SD |
|---|---|---|---|
| **Health utility estimate** | | | |
| Year 1 CP to Year 2 CP | | | |
| Year 1 | 25 | 0.365 | 0.308 |
| Year 2 | 25 | 0.326 | 0.328 |
| Year 1 CP to year 2 non-CP | | | |
| Year 1 | 17 | 0.524 | 0.276 |
| Year 2 | 19 | 0.619 | 0.216 |
| Year 1 non-CP to year 2 non-CP | | | |
| Year 1 | 374 | 0.811 | 0.189 |
| Year 2 | 368 | 0.826 | 0.212 |
| Year 1 non-CP to year 2 CP | | | |
| Year 1 | 21 | 0.604 | 0.175 |
| Year 2 | 21 | 0.461 | 0.317 |
| **OKS-PS** | | | |
| Year 1 CP to year 2 CP | | | |
| Year 1 | 25 | 9.4 | 3.73 |
| Year 2 | 25 | 9.4 | 3.93 |
| Year 1 CP to year 2 non-CP | | | |
| Year 1 | 19 | 11.2 | 3.08 |
| Year 2 | 19 | 18.7 | 2.75 |
| Year 1 non-CP to year 2 non-CP | | | |
| Year 1 | 382 | 24.3 | 3.63 |
| Year 2 | 382 | 24.9 | 3.56 |
| Year 1 non-CP to year 2 CP | | | |
| Year 1 | 22 | 17.2 | 2.48 |
| Year 2 | 22 | 11.7 | 2.34 |

OKS-PS, Oxford Knee Score Pain Subscale.

(average healthcare costs: £1800 for CP, £500 for non-CP). The main driver for the difference was readmission to the same hospital. Although the questionnaires used in COASt did not record the reason for readmissions, they were likely due to reoperations.

Two-thirds (64.7%) of participants with CP recovered and were no longer in CP within 5 years after surgery. This result suggests that CP after TKR is not a permanent condition and can improve with time as patients access routine care. There can be many reasons for these changes. They could be linked to individual characteristics, day-to-day life, or the severity of the original condition. Qualitative work from the STAR (Support and Treatment After Replacement) programme suggests that acceptance and self-management could potentially play a role, although this is an area that needs more research.[39] Another speculative hypothesis is that the fact that the CP group had more severe pain and reduced function prior to surgery may indicate that they were more deconditioned prior to surgery. This could explain the longer time to recovery, which would be supported by the

fact that they had higher number of physiotherapy visits in the postoperative period. Furthermore, it may be that some individuals learn to live with the pain overtime and choose self-management approaches.[39][40]

Most of those who recovered from CP did so during the second year after surgery. One possible explanation for the prolonged recovery period from postsurgical pain may be in relation to the different pain mechanisms which are involved in OA.[41] Although the current study does not include any measures of centrally mediated pain, the presence of central sensitisation has been previously demonstrated in patients with OA awaiting knee and hip arthroplasty and is also likely to be associated with worse outcome following arthroplasty.[42–44] While preliminary data suggest that some features of altered brain morphology associated with pain in osteoarthritis are potentially reversible within the first year after surgery,[45] the nature and timing of any potential resolution of central sensitisation after surgery requires further investigation. Studies which have used questionnaire-based screening methods to identify features of central sensitisation in the preoperative and postoperative periods suggest it is likely to persist at least up to 2 years after surgery.[46–49]

Only 4.4% of participants reported OKS-PS scores that indicated CP for all 5 years after surgery. Brander et al[50] also found most patients with heightened, unexplained pain 1 year after knee replacement demonstrated subsequent improvement over several years. They found that depression was an important determinant of long-term outcomes.

One-third (30.9%) of participants reported fluctuating OPS-PS scores that classified them as experiencing CP at only certain points during the study period. Relief from CP may not always be permanent, which is consistent with the variability in temporal fluctuation in knee pain seen without surgical intervention.[51] Although most participants eventually recovered from CP and made associated gains in HRQL, those who experienced CP 1 year after surgery did not reach the same level of health utility 5 years after surgery as without CP (0.659 vs 0.787, respectively). To our knowledge, this is the first study applying a CP criterion to examine whether and when people with CP after TKR recover.

Participants in CP after the first year after surgery who recovered by the second year improved their health utility more than those who stayed in CP, but not as much as those who were not in CP in either year. Improved OKS-PS is a sign of less pain. As it is likely associated with improved mobility, self-care and conduct of usual activities, it would be expected to lead to improved EQ-5D scores. However, participants switching CP groups also reported changes in use of healthcare resources that separated them from the CP group and brought them closer, although not directly in line with, participants without CP. These findings suggest progressive improvement as people come out of CP. The OKS-PS cut-off used to identify CP may therefore be sensitive enough to identify distinct groups as they transition between CP categories.

Our results suggest that under current clinical practice, many patients leave CP over time by gradually improving. Identifying these patients early will allow clinicians to provide them with tailored support to make that transition as quickly as possible. Better understanding which patients remain in CP or only recover after many years of struggling with pain will also help clinicians to design interventions to improve and manage these patients' pain, allow them to continue with their lives with the least interference possible.

This study has several potential limitations. As our analyses used data collected from patients recruited from two hospitals in Southampton and Oxford, the findings are not necessarily generalisable to the rest of the country. However, the participants' health gains after TKR agreed with those reported by the NHS PROMs programme,[37] which invites all patients undergoing a knee replacement funded by the NHS in England to take part, suggesting a degree of generalisability. It is also possible that contralateral replacements and revisions, although unlikely, might have influenced the findings.

Our categorisation of CP was made using an OKS-PS cut-off score. As pain is a complex construct, it might not be appropriate to dichotomise it via a threshold on an instrument designed to measure improvement after knee replacement, rather than capturing the diverse dimensions of the pain experience. However, the use of this cut-off point to identify people with and without CP effectively distinguished between groups with significantly different health outcomes (including HRQL), resource use and healthcare costs,[4] which can impact future guidelines for clinical care for people with CP. It should also be highlighted that CP postsurgery will not be the only factor than influences the HRQL, healthcare resource use and costs of an individuals who has undergone a TKR and there may be additional causes of these changes.

The questionnaires used in COASt resulted in many study limitations, only some of which could be mitigated against. Many follow-up questionnaires were not returned, which generated an important level of missing data. It is difficult to know the direction of the impact this may have had as participants may not return questionnaires for multiple reasons, including both being very dissatisfied with their surgery or doing so well they could not be bothered to respond. We addressed this issue by applying multiple imputation to reduce bias and improve our study's power.[17][52] We also analysed the observed and missing data at 5-year follow-up and did not find a significant difference between the groups. Previous research has shown that assuming that missing data are either missing at random or missing completely at random, even with lost to follow-up of 60%, does not necessarily lead to significant bias of the results.[53] Research has also shown that lost to follow-up can be predicted by measures of pain and functioning, which supports our findings that dimensions of EQ-5D and OKS-PS are associated with missing data.[54]

The questionnaires did not allow us to identify specific interventions received by participants after TKR, only unspecified hospital readmissions. We were, therefore, unable to identify whether the improvements observed were due to any particular interventions or treatments. Moreover, lacking specific intervention details was a limitation to estimate costs to the NHS, but we addressed this by using average unit costs weighted by their relative use in the NHS.[14]

As the resource-use questionnaires asked patients to report events that had occurred over the previous 12 months, they were potentially subject to recall bias. However, we are confident that most hospitalisations are sufficiently significant events for participants to have recalled them all. However, this may not be the case for less significant events such as a routine GP visit. Due to the potential bias from the self-reported resource use and the influence from factors not available for this study, the costs estimation should be used with caution and considered a broad estimate.

The questionnaires did not ask participants about informal care, productivity losses, or their use of privately funded healthcare other than physiotherapists. Although these issues are relevant for TKR patients, excluding this information was consistent with the perspective adopted for this analysis, that is, that of the healthcare payer (NHS).

This study's main strength is the rich longitudinal dataset used. Unlike the national PROMs programme, which only collects PROMs before and 6 months after surgery, COASt collected data for 5 years after TKR. It also offered data on both PROMs and healthcare resource use, allowing us to explore the patterns in both.

Further research building on our findings is warranted. Identifying how best to assess presurgical and postsurgical factors may help to provide more detailed understanding of variation in and potential predictors of CP and the likelihood and speed of recovery. We believe that including measures such as neuropathic pain, catastrophising, sleep disturbance, anxiety, depression, pain medication intake, self-prescription and physical activity, among others, would help build a comprehensive, informative picture. In the meantime, clinicians can reassure patients that, though many fluctuate in their levels of pain, CP after TKR is generally not permanent and that improvement over the 5 year after their primary is likely for many, if not most.

## CONCLUSION

We identified a significant, rapid improvement in HRQL for people undergoing a TKR. This overall trend, however, appears to hide the slow and gradual improvement experienced by people with CP 1 year after their primary. They differed from those not in CP in terms of their progression of pain and function, HRQL, healthcare resource use and costs. Our study suggests that the majority of people with CP after the first year eventually recovered by the fifth year after TKR, reducing their need for healthcare. Further research is needed to understand the reasons for this difference in progression as well as who is most likely to develop CP after TKR and how best to support patients in their long-term recovery, benefiting patients, their families and healthcare systems.

**Acknowledgements** We thank the participants of the COASt study in Southampton General Hospital and Nuffield Orthopaedic Centre in Oxford as well as all healthcare professionals and researchers involved. As a study conducted as part of the STAR programme, we acknowledge all members and affiliated members of the team: Aideen Ahern, Ashley Blom, Amanda Burston, Jane Dennis, Kirsty Garfield, Athene Lane, Fiona MacKichan, Sian Noble, Andrew Moore, Tim Peters, Emily Sanderson, Joanne Simon, Jodi Taylor, Paul Dieppe, Chris Eccleston, Nick Ambler, Wendy Bertram, Susan Bridgewater, Nick Howells, Leigh Morrison, Gemma Munkenbeck, Candida McCabe, Andrew Toms, Rowenna Stroud, Kate Button, Simon White, Andrew Price, David Walsh, Julie Bruce, Stewart Long, Joanne Adams, Ben Burston, Vikran Desai, Tim Board, Colin Esler, Michael Parry, and the Patient Experience Partnership in Research (PEP-R) at the University of Bristol.

**Contributors** RP-V led the design of the work with substantial contributions from SC, SK, AS and RG-H. SC, SK, AS, AD, MSS, AJ, NKA, ADB, VW, RG-H and RP-V contributed to the planning of the study. SC, SK and RP-V completed the data extraction criteria and AD curated and extracted the data. SC and SK undertook the data cleaning, SC and MSS conducted the multiple imputation, and SC did the analyses under the supervision of RP-V. SC, SK, MSS and RP-V led the reporting of the methods and findings. RP-V met with patients to gather PPI feedback. SC, SK, AS, AD, MSS, AJ, NKA, ADB, VW, RG-H and RP-V contributed to the interpretation of data. SC and RP-V wrote the original draft and SK, AS, AD, MSS, AJ, NKA, ADB, VW and RG-H made substantial contributions to subsequent versions and revised it critically for important intellectual content. SC, SK, AS, AD, MSS, AJ, NKA, ADB, VW, RG-H and RP-V approved the final version of the manuscript and agree to be accountable for all aspects of the work by ensuring that questions related to the accuracy or integrity of any of its parts are appropriately investigated and resolved. RP-V acted as guarantor.

**Funding** The STAR Programme was funded by the National Institute for Health Research (NIHR) (Programme Grant for Applied Research (Grant Reference Number RP-PG-0613-20001)) and supported by the NIHR Biomedical Research Centre at University Hospitals Bristol and Weston NHS Foundation Trust and the University of Bristol. We acknowledge English language editing by Dr Jennifer A. de Beyer of the Centre for Statistics in Medicine, University of Oxford.

**Disclaimer** The views expressed are those of the authors and not necessarily those of the NIHR or the Department of Health and Social Care.

**Competing interests** NKA reports consultancy fees outside the scope of this work.

**Patient and public involvement** Patients and/or the public were involved in the design, or conduct, or reporting, or dissemination plans of this research. Refer to the Methods section for further details.

**Patient consent for publication** Not applicable.

**Ethics approval** The COASt study was approved by the Oxford Research Ethics Committee A (reference number 10/H0604/91). Research complied with the Helsinki Declaration.

**Provenance and peer review** Not commissioned; externally peer reviewed.

**Data availability statement** No data are available.

**ORCID iDs**
Sophie Cole http://orcid.org/0000-0002-5853-2427
Anushka Soni http://orcid.org/0000-0002-7831-4208
Maria T Sanchez-Santos http://orcid.org/0000-0003-1908-8623
Andrew David Beswick http://orcid.org/0000-0002-7032-7514
Rachael Gooberman-Hill http://orcid.org/0000-0003-3353-2882
Rafael Pinedo-Villanueva http://orcid.org/0000-0002-4723-5128

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
