## [Reviewer comments · BMJ Open]

ARTICLE DETAILS

TITLE (PROVISIONAL)	Progression of chronic pain and associated health-related quality of life and healthcare resource use over 5 years after total knee replacement: evidence from a cohort study
AUTHORS	Pinedo-Villanueva, Rafael; Cole, Sophie; Kolovos, Spyros; Soni, Anushka; Delmestri, Antonella; Sanchez Santos, Maria; Judge, Andrew; Arden, Nigel; Beswick, Andrew; Wylde, Vikki; Goberman-Hill, Rachael

VERSION 1 – REVIEW

REVIEWER	Larsen, Jesper Bie Aalborg Universitet, Department of Health Science and Technology
REVIEW RETURNED	22-Nov-2021

GENERAL COMMENTS	Comments to BMJ Open submission ID bmjopen-2021-058044 The manuscript investigated pain and function, health-related quality of life (HRQL), and healthcare resource use over 5 years after total knee replacement (TKR). Chronic pain was defined using the Oxford Knee Score pain subscale (OKS-PS) 1-year post-surgery to discriminate between those categorized as having chronic pain (CP) and those that didn't have CP. Chronic pain was reported in 12.7% of the patients at 1-year post-surgery. Between-group comparison revealed that the CP group had worse pain, function, and HRQL pre- and post-surgery than the non-CP group. The authors have clearly invested a lot of work into the cohort study and long-term follow-up studies are requiring. Unfortunately, these studies often come with poor adherence and large dropout, which is also the case in the present study. The authors openly and clearly report the drop-out rates and their handling of these issues from a statistical point-of-view. I lack the appropriate statistical insight to be able to review whether imputation methods can account for such a massive drop-out rate, so I advise that a statistician should review the statistical analysis. I have other major concerns that I suggest the authors to consider, which are related to the lack of reporting the statistical tests used for between-group comparisons and related to the definition of whether CP is present or not. Below, I have given my considerations for the manuscript. General comments The manuscript is comprised of a lot of data, various analysis and consists of 14 tables and 8 figures (including supplementary material). I struggle to figure out what the main focus of the manuscript is, e.g., what is the main outcome or analysis of
---

	interest? I think the author should specify this and focus the manuscript on this. If everything is important then nothing is important! As of now I think you present a lot of data without explaining to the reader what the main thing is to focus on. The authors rationale is that pain will cause the changes in HRQL and healthcare use and costs, but I think it should be acknowledged that other factors could very well contribute to the outcomes. I don't think you can conclude that pain alone is the cause of the follow-up assessments. For instance, deficits in range of motion following TKR is a well-known challenge that often isn't perceived as painful but influences the patients HRQL and might even lead to surgical procedures which will inflate the costs. As of now your rationale doesn't take such factors into account. Definition of CP is the key element, and the authors use a cut-off for the OKS-PS to establish that patients experience CP. I can't help wondering that when the pain seems to fluctuate (as the data shows) how prone is the pain assessment to patients normally being pain-free, but presently is experiencing a pain flare-up – or the opposite? Another issue that should be reported and discussed is that those (or at least some of them) classified as non-CP seems to experience pain as well, giving that the OKS-PS score does not indicate no pain. I think the authors should elaborate further on why some patients with low level of pain can't be considered as experiencing CP and those with higher levels of pain is considered with CP. Through-out the manuscript you switch between the word quality of life and health-related quality of life. Suggest being consistent. Title: Consider replacing "quality of life" with "health-related quality of life" since this is the terminology used through-out the manuscript. Abstract: Suggest specifying what the STAR program is. Please specify the cut-off in OKS-PS used to determine whether CP or not. Introduction: Page 6, line 8: Suggest inserting the word "knee" before osteoarthritis Page 6, line 15-16: Suggest specifying that this is related to CP following TKR. Partly, because this is the topic of the present manuscript, partly, because there are studies available looking into the trajectories for e.g., knee OA. Page 6, line 37-38: I think the sentence of "...helping clinicians to design effective interventions to treat or manage CP" is taken a bit too far. How is it that knowing the pain trajectory will lead to improved treatment? It might explain the trajectories, but we still don't have any effective treatments available. Line 42-43: I can't help wondering if the sentence "assessing clinical and cost-effectiveness is a critical step in informing decision-making about the wider implementation of new interventions" is out of scope? I don't think the objective of the manuscript is to evaluate whether TKR should be considered as treatment or not? Line 49-50: The phrase "HRQL is affected by the severity of pain and its consequences and is expected to vary as people's pain improves, worsens, or fluctuates" is a bit simplified as many other factors influence HRQL. Suggest to either back the statement with a reference and/or rephrase so it includes that other things influence HRQL (e.g., psychosocial factors). Page 7, line 6: Maybe it is my lack of understanding, but how can cost-effectiveness be assessed when you split patients with the
--	--

	same intervention (TKR) into two groups? Would this not only allow you to report differences in healthcare use, but not to evaluate the cost-effectiveness? Page 7, line 10-11: I think it should be clarified why pain at 1-year post-operative was the cut-off chosen? Previously you stated that pain is chronic by definition after 3 months, so I suggest explaining why it was relevant to consider at 1-year post-operatively (which I agree makes sense). Patients and methods: Overall, I think a large part of the sections COASt and patient and public involvement could be shortened down and instead refer to the original manuscripts/protocols. At 5-years follow-up I think it is of minor importance to clearly specify how the study was conducted as this has already been published. However, some information regarding the population and the inclusion/exclusion criteria are lacking; Was there limitations regarding age, BMI, or cause of joint replacement surgery? Could patient have bilateral surgery and if so, how did you make sure that the same knee was evaluated over time? Was revision surgery tracked and taken into account? If patients had a TKR in the opposite knee how was that handled, e.g. exclusion or could the same patients be in the dataset twice? Page 8, line 13-14: As previously stated, I lack the statistical insight regarding imputation methods, but I find it difficult to follow that main variable of interest (whether patients had CP or not a 1-year) is based on imputed data and not actual observations. Page 9, line 57: When assessing healthcare use based on self-reporting using an unvalidated (I assume) questionnaire over the last year your results are prone to bias (which the authors also acknowledge as a limitation). Furthermore, you have no knowledge whether there are comorbidities associated with CP, which could influence the healthcare, e.g. the presence of depression could influence your pain perception and cause you to consult the healthcare system. Therefore, I would be careful when interpreting the findings, acknowledging that this is probably a very “raw” results, which could very well be influenced by a lot of other factors that you haven’t taken into account. Page 10, line 8-9: Given my concerns mentioned above, I struggle to see the real value of the analysis of the costs. Not only is the costs analysis based on the “raw” findings, but it also doesn’t take other healthcare related costs into consideration such as expenditures for medication, productivity loss. I get that the authors would like to illustrate that patients with or without CP after TKR have different trajectories and healthcare use, but does it add anything to this scope when calculating the costs? I am not sure and suggest considering whether this part is truly adding value to the manuscripts scope, especially considering the massive amount of analysis already included. Also, I can’t help wondering whether costs are a relevant outcome measure when evaluating CP after TKR. A patient undergoing surgery for range of motion deficits or infection would probably present themselves with high cost, whereas patients with “unexplained” chronic pain won’t undergo surgery and won’t get a standardized treatment (because we don’t have evidence-based treatments for chronic pain following TKR), so they are likely to present lower costs. Analysis: Overall, this section should specify which statistical tests were used to evaluate the between-group comparisons and clarify which statistical measures will be presented, e.g., p-values, 95% confidence intervals etc.
--	---

	Page 11, line 6: Suggest specifying what is meant by “outcomes” Page 11, line 13-14: When stratifying patients into with or without CP at 1-year what was then the rationale for looking at pre-operative data as well? Could the authors clarify why this was important to take into consideration as well? Results Overall, I think the manuscript would benefit from shortening down on the analysis and presented data as there is a massive amount of data presented. E.g., table 1 presents data for mean age and age categories, and IMD data for 10 categories. I would suggest that the author considers what is really necessary to report when the overall conclusion is that no differences between group are present. Also consider if it makes more sense to report SD instead of ranges when reporting mean values. Table 2 also include a lot of data, making it difficult to figure out what the most important findings are? Page 12, line 18-22: How did you evaluate that the CP group had worse outcomes? Which statistical analysis were this based on? I also suggest reporting exact p-values and 95% confidence intervals for the differences because of the explorative design. Given the differences in baseline characteristics for sex and BMI did you then adjust for this in the between-groups analysis? Page 15, line 4: Suggest specifying what is meant by “outcomes” Page 15, line 5-29: I don’t think this section should be part of the results. It seems to be related to validation of the statistical methods and not actual results related to the focus of the manuscript? Consider placing in the “analysis” section. Page 15, line 30-31: How was the reported difference calculated and could the authors provide some estimates for the differences? Page 15, line 40: When writing “significantly” I think the authors should present the estimates to back up this statement. Page 16, line 47-48: What is A&E an abbreviation for? Page 17, line 23-41: From my perspective this is the most interesting / novel finding in the study. The authors could consider focusing and build the manuscript around this. Page 17, line 45: I struggle to see the relevance of an analysis between 1- and 2-years follow-up? What was the rationale for this specific timepoint and no other timepoints? Discussion Overall, the challenge for the discussion is the fact that you have observed some changes in CP rates, HRQL etc. over time, but you have no idea of WHY these changes occurred – which is the interesting part. Though it will be a bit speculative I suggest that the authors try to explain what could be behind the changes, i.e., why their TKR population shows good results over time. Could it be because the rehabilitation pathways following TKR or did the patients learn to live with the pain? Your research-group has vast experience with this population including qualitative data from these patients and could perhaps make an attempt to provide answers. Page 20, line 33: The introduction of improvement of centrally mediated pain mechanisms as explanation is highly speculative. The present study has not investigated any central pain mechanisms and whether central pain mechanisms can normalize following replacement surgery remains debatable. Page 24, line 3: Is the sentence “clinicians can reassure patients that CP after TKR is generally not permanent and that improvement is likely for many, if not most” true giving the findings? 12% did report CP and your data shows that patients fluctuate in and out of the CP category. I think the authors should
--	--

	reconsider if their data have enough validity to make such a statement Conclusion Again, I would be cautious to state that most patients recover from TKR over time. It might be correct, but giving the limitations of massive drop-out, CP categorization based on a cut-off score from the OKS-PS, future research could very well report findings in contrast with the present results. I think the findings is more explorative than definitive and suggest reporting as such. Acknowledgements A minor comment, but I find somewhat strange that several of the authors are also listed in the acknowledgement section? For me, the acknowledgement section is about acknowledging those that participating or contributed to the study, but not enough to qualify as authors.
--	---

REVIEWER	Ahmed, Bawan University of Greenwich
REVIEW RETURNED	01-Feb-2022

GENERAL COMMENTS	Thank you for your work, I enjoyed reading your paper, to further enrich this paper I suggest minor amendments, below you will find my comments:  1. Line 8, page 6: I suggest changing "osteoarthritis" to ""end-stage knee osteoarthritis" 2. Lines 15-16, page 6: "There is currently limited understanding of how chronic pain (CP)" It is unclear to the reader whether or not CP is knee/post surgical specific or not so I suggest using "Knee OA associated CP, or post surgical CP?" This comment also applies to the rest of the paper. 3. It would be really useful and its important if you can add further patients characteristics such as presence or absence of deformities (fixed flexion and varus) both pre and post operatively as a surgical variable this is in addition to ACL (anterior cruciate ligament).
---

VERSION 1 – AUTHOR RESPONSE

Reviewer 1	
The manuscript is comprised of a lot of data, various analysis and consists of 14 tables and 8 figures (including supplementary material). I struggle to figure out what the main focus of the manuscript is, e.g., what is the main outcome or analysis of interest? I think the author should specify this and focus the manuscript on this. If everything is important then nothing is important! As of now I think you present a lot of data without explaining to the reader what the main thing is to focus on.	We agree that we have included considerable data displays and analyses and that this may mean that the central focus of the manuscript is not as clear as it could be. We have ensured that the abstract clearly reflects the main aim of the study and have made edits to ensure that this is the case We have therefore made a change to the title to reflect that the main aim is to describe how outcomes evolve over five years for those with and without chronic pain at one year after surgery. We have also refined introduction so readers can see the focus more clearly.

The authors rationale is that pain will cause the changes in HRQL and healthcare use and costs, but I think it should be acknowledged that other factors could very well contribute to the outcomes. I don't think you can conclude that pain alone is the cause of the follow-up assessments. For instance, deficits in range of motion following TKR is a well-known challenge that often isn't perceived as painful but influences the patients HRQL and might even lead to surgical procedures which will inflate the costs. As of now your rationale doesn't take such factors into account.	This is a helpful comment, and we agree that pain is not the only cause of changes in HRQL and demand for health care. To make this clearer we have added text in the introduction and discussion to acknowledge that pain is one of several factors that can influence HRQL and healthcare use and costs.
Definition of CP is the key element, and the authors use a cut-off for the OKS-PS to establish that patients experience CP. I can't help wondering that when the pain seems to fluctuate (as the data shows) how prone is the pain assessment to patients normally being pain-free, but presently is experiencing a pain flare-up – or the opposite?	We agree with the reviewer that the definition of CP is important in relation to fluctuation. Our discussion included a brief paragraph about the limitations of using a chronic pain threshold. To help the readers navigate the impact of such limitations, we discussed the sensitivity of the OKS-PS cut-off by examining the average OKS-PS and health utility estimates of those patients who fluctuated in and out of CP between years 1 and 2 to see whether those fluctuations were a marked improvement or, as suggested, perhaps the experience of a flare-up. More in-depth exploration of this would perhaps require taking several OKS-PS measurements throughout the year which certainly could be interesting but fall outside the scope of this study.
Another issue that should be reported and discussed is that those (or at least some of them) classified as non-CP seems to experience pain as well, giving that the OKS-PS score does not indicate no pain. I think the authors should elaborate further on why some patients with low level of pain can't be considered as experiencing CP and those with higher levels of pain is considered with CP.	The threshold is meant to distinguish, as labelled, between those with and without chronic pain, not with and without pain and to identify patients with a level of pain that would negatively affect quality of life. It is to be expected that some patients categorised as non-CP would experience some pain, but the broader association between chronic pain and quality of life as well as demand for healthcare services remain. So that this is clear to readers, we have elaborated further on this in the methods section.
Through-out the manuscript you switch between the word quality of life and health-related quality of life. Suggest being consistent.	We have edited the wording throughout the manuscript so that we refer to "health-related quality of life" consistently.
Title: Consider replacing "quality of life" with "health-related quality of life" since this is the terminology used through-out the manuscript.	We have updated the title accordingly.

Abstract: Suggest specifying what the STAR program is.	We have now specified what the STAR programme is in the abstract.
Abstract: Please specify the cut-off in OKS-PS used to determine whether CP or not.	We have added specification of the cut-off for CP in the abstract.
Introduction: Page 6, line 8: Suggest inserting the word “knee” before osteoarthritis	We have added “knee” before “osteoarthritis” as suggested.
Introduction: Page 6, line 15-16: Suggest specifying that this is related to CP following TKR. Partly, because this is the topic of the present manuscript, partly, because there are studies available looking into the trajectories for e.g., knee OA.	We have updated the sentence to specify that this is CP following TKR.
Introduction: Page 6, line 37-38: I think the sentence of “...helping clinicians to design effective interventions to treat or manage CP” is taken a bit too far. How is it that knowing the pain trajectory will lead to improved treatment? It might explain the trajectories, but we still don’t have any effective treatments available.	We agree that this phrasing is not clear enough and that there is overreach. We have rephrased the sentence to “helping clinicians gain greater insight into the condition, hopefully contributing to finding ways to treat patients more effectively” to address this.
Introduction: Line 42-43: I can’t help wondering if the sentence “assessing clinical and cost-effectiveness is a critical step in informing decision-making about the wider implementation of new interventions” is out of scope? I don’t think the objective of the manuscript is to evaluate whether TKR should be considered as treatment or not?	Thank you for noting this. Our statement was meant to refer to how to assess future potential treatments for chronic pain (not the TKR itself). We appreciate the ambiguity in the statement and have rephrased it so that we are expressing this with greater clarity.
Introduction: Line 49-50: The phrase “HRQL is affected by the severity of pain and its consequences and is expected to vary as people’s pain improves, worsens, or fluctuates” is a bit simplified as many other factors influence HRQL. Suggest to either back the statement with a reference and/or rephrase so it includes that other things influence HRQL (e.g., psychosocial factors).	We have rephrased the statement to highlight that there are other factors that influence HRQL: “HRQL is affected by a number of factors including the severity of pain and its consequences and is expected to vary as people’s pain improves, worsens, or fluctuates”.
Introduction: Page 7, line 6: Maybe it is my lack of understanding, but how can cost-effectiveness be assessed when you split patients with the same intervention (TKR) into two groups? Would this not only allow you to report differences in healthcare use, but not to evaluate the cost-effectiveness?	We apologise for the lack of clarity, we did not mean to imply we had completed cost-effectiveness analysis but rather our study may aid and enable future cost-effectiveness analysis. We have rephrased the sentence to make this clearer.
Introduction: Page 7, line 10-11: I think it should be clarified why pain at 1-year post-operative was the cut-off chosen? Previously you stated that pain is chronic by definition after 3 months, so I suggest explaining why it was relevant to consider at 1-year post-operatively (which I agree makes sense).	We have included an additional sentence to explain and clarify this: “We considered CP 1-year post-operatively to give patients sufficient time to recover from the operation and ensure that any pain reported was not due to the operation itself.”.

Patients and methods: Overall, I think a large part of the sections COASt and patient and public involvement could be shortened down and instead refer to the original manuscripts/protocols. At 5-years follow-up I think it is of minor importance to clearly specify how the study was conducted as this has already been published.	Detail of the patient involvement related to this work has not been previously published and we feel it is important to report it here. However, we have shortened the PPI section so that it is more concise but still contains key details.
Patients and methods: However, some information regarding the population and the inclusion/exclusion criteria are lacking; Was there limitations regarding age, BMI, or cause of joint replacement surgery? Could patient have bilateral surgery and if so, how did you make sure that the same knee was evaluated over time? Was revision surgery tracked and taken into account? If patients had a TKR in the opposite knee how was that handled, e.g. exclusion or could the same patients be in the dataset twice?	We have added further information about the population as well as inclusion and exclusion criteria in the methods section. As the lack of information about contralateral surgeries and revisions might have impacted findings, we have also added a sentence about this in the discussion.
Patients and methods: Page 8, line 13-14: As previously stated, I lack the statistical insight regarding imputation methods, but I find it difficult to follow that main variable of interest (whether patients had CP or not a 1-year) is based on imputed data and not actual observations.	Imputation of missing data is common practice when data are missing, even (and perhaps particularly) for the main variable of interest. Best practice methods were followed to conduct the multiple imputation using chained equations, which has limitations as highlighted in the discussion, but allows the analysis of data from patients who would have otherwise been removed (and hence likely introducing bias) if we had conducted a complete-case analysis instead. We have added a sentence in the methods to address this: “. Missing data and attrition are an important concern for longitudinal studies. It has been argued that the imputation of missing covariates data in medical research is always better than the complete case. Excluding subjects with missing values leads to a reduction in the sample study size and may diminish the predictive power of the working statistical model.”.
Patients and methods: Page 9, line 57: When assessing healthcare use based on self-reporting using an unvalidated (I assume) questionnaire over the last year your results are prone to bias (which the authors also acknowledge as a limitation). Furthermore, you have no knowledge whether there are comorbidities associated with CP, which could influence the healthcare, e.g. the presence of depression could influence your pain perception and cause you to consult the	We completely agree that self-reported questionnaires may introduce bias and there is potential for other comorbidities to be associated with CP which haven't been included and have further acknowledged this limitation in the discussion.

healthcare system. Therefore, I would be careful when interpreting the findings, acknowledging that this is probably a very “raw” results, which could very well be influenced by a lot of other factors that you haven’t taken into account.	
Patients and methods: Page 10, line 8-9: Given my concerns mentioned above, I struggle to see the real value of the analysis of the costs. Not only is the costs analysis based on the “raw” findings, but it also doesn’t take other healthcare related costs into consideration such as expenditures for medication, productivity loss. I get that the authors would like to illustrate that patients with or without CP after TKR have different trajectories and healthcare use, but does it add anything to this scope when calculating the costs? I am not sure and suggest considering whether this part is truly adding value to the manuscripts scope, especially considering the massive amount of analysis already included. Also, I can’t help wondering whether costs are a relevant outcome measure when evaluating CP after TKR. A patient undergoing surgery for range of motion deficits or infection would probably present themselves with high cost, whereas patients with “unexplained” chronic pain won’t undergo surgery and won’t get a standardized treatment (because we don’t have evidence-based treatments for chronic pain following TKR), sthey are likely to present lower costs.	We agree that the cost estimates are ‘raw’, however given the lack of any reference to even minimum levels of healthcare costs associated with chronic pain, we believe it is of great value to readers to have a reference point, albeit with important limitations. We believe they are an important part of the characterisation of chronic pain following a TKR. From our analysis it is clear that those in CP (understandably) report higher costs than those in without CP and being able to report this alongside the differences in HRQL we think is worthwhile.
Analysis: Overall, this section should specify which statistical tests were used to evaluate the between-group comparisons and clarify which statistical measures will be presented, e.g., p-values, 95% confidence intervals etc.	We have now specified this in the analysis section. We have also added a sentence in the analysis to mention that we calculated confidence intervals for health utility and OKS-PS.
Analysis: Page 11, line 6: Suggest specifying what is meant by “outcomes”	We have added specification of the outcomes (HRQL and pain).
Analysis: Page 11, line 13-14: When stratifying patients into with or without CP at 1-year what was then the rationale for looking at pre-operative data as well? Could the authors clarify why this was important to take into consideration as well?	This is an interesting point and we have now clarified this in the analysis section by adding the following sentence: “We considered pre-surgery as well as post-surgery to investigate whether those with CP post-surgery report distinct differences even prior to surgery.”.
Results: Overall, I think the manuscript would benefit from shortening down on the analysis and presented data as there is a massive amount of data presented. E.g., table 1 presents data for mean age and age	We have combined items in Table 1 to substantially reduce its length, especially IMD which now has four categories instead of 10. We report SD together with the mean for

categories, and IMD data for 10 categories. I would suggest that the author considers what is really necessary to report when the overall conclusion is that no differences between group are present. Also consider if it makes more sense to report SD instead of ranges when reporting mean values. Table 2 also include a lot of data, making it difficult to figure out what the most important findings are?	BMI but have chosen to show the range when reporting age as we believe it is informative.
Results: Page 12, line 18-22: How did you evaluate that the CP group had worse outcomes? Which statistical analysis were this based on? I also suggest reporting exact p-values and 95% confidence intervals for the differences because of the explorative design. Given the differences in baseline characteristics for sex and BMI did you then adjust for this in the between-groups analysis?	We have now added p-values from T-tests comparing outcome scores between the CP and Non-CP groups in Table 2, showing that they are statistically significantly different (p-value \leq 0.001). We did not adjust for gender and BMI in the between group analysis. However, we did run linear regression analysis on the outcomes with CP group, gender, and BMI as covariates. We observed similar statistically significant differences in the health outcomes between the CP groups whilst adjusting for the two potential cofounders. We have now reported this at the end of the first paragraph of the results.
Results: Page 15, line 4: Suggest specifying what is meant by “outcomes”	This has now been specified as HRQL and pain outcomes.
Results: Page 15, line 5-29: I don't think this section should be part of the results. It seems to be related to validation of the statistical methods and not actual results related to the focus of the manuscript? Consider placing in the “analysis” section.	We understand the reviewer's comment and suggestion but as this section reports on the findings of missing data, which is important for the interpretation of results, we believe it is best suited to be reported as part of the Results and not within the Methods section.
Results: Page 15, line 30-31: How was the reported difference calculated and could the authors provide some estimates for the differences?	Mean health utility estimates are provided in the sentence that follows for each of the groups. We expanded the sentence to highlight that the confidence intervals overlap: “The CP and non-CP groups had different mean health utility estimate scores before surgery (Figure 1) although their confidence intervals overlapped (Electronic Supplementary Material Table A.5)”.
Results: Page 15, line 40: When writing “significantly” I think the authors should present the estimates to back up this statement.	We have added the pre-operative score within the sentence to show the large improvement at year 1.
Results: Page 16, line 47-48: What is A&E an abbreviation for?	The abbreviation has been removed from the text and replaced with the full wording.
Results: Page 17, line 23-41: From my perspective this is the most interesting / novel finding in the study. The authors could consider focusing and build the manuscript around this.	We too believe the findings about the evolution of resource use to be novel, but as suggested by the reviewer we also had to maintain the focus of the manuscript centred on the main aim, which was the progression

	of chronic pain, and secondly its associated resource use and costs. We report and later discuss the resource use findings at length and believe there is now a good balance of those findings in the context of the progression of outcomes and, to a lesser extent, costs.
Results: Page 17, line 45: I struggle to see the relevance of an analysis between 1- and 2-years follow-up? What was the rationale for this specific timepoint and no other timepoints?	This analysis was important to explore whether those who are moving in and out of CP are doing so because of a distinct deterioration (or improvement) or whether they had just fluctuated slightly above or below the chronic pain cut-off (OKS-PS\leq14). This was also highlighted in the response to a previous comment by the reviewer where the possibility of small changes in the OKS-PS might lead to changes in the labelling of CP even if perhaps the patients were not experiencing an important change in quality of life. We do explain in the Methods section, under “Analysis” (second to last paragraph), the purpose of this investigation.
Discussion: Overall, the challenge for the discussion is the fact that you have observed some changes in CP rates, HRQL etc. over time, but you have no idea of WHY these changes occurred – which is the interesting part. Though it will be a bit speculative I suggest that the authors try to explain what could be behind the changes, i.e., why their TKR population shows good results over time. Could it be because the rehabilitation pathways following TKR or did the patients learn to live with the pain? Your research-group has vast experience with this population including qualitative data from these patients and could perhaps make an attempt to provide answers.	We have now included some potential reasons for these improvements in the discussion. Thank you, reasons for change over time are fascinating and important. We appreciate the reviewer’s confidence in our research team’s expertise but we have to admit that we are concerned that speculation about reasons for fluctuation would be overstepping the mark for this publication. Having said this we would be comfortable citing our recent publication about the STAR care pathway and its value to patients with CP, and we have made additions to reflect this in the discussion of these findings.
Discussion: Page 20, line 33: The introduction of improvement of centrally mediated pain mechanisms as explanation is highly speculative. The present study has not investigated any central pain mechanisms and whether central pain mechanisms can normalize following replacement surgery remains debatable.	Thank you for highlighting this oversight. We have edited the text to make it clear that this is a possible explanation which requires further investigation. We have also updated the references to provide more details about the currently available evidence regarding the impact of central pain mechanisms on the recovery from arthroplasty. Furthermore, in combination with the other explanations that have been described, in response to the reviewer’s previous comment, the emphasis on central pain mechanisms as a proposed explanation has been reduced.

Discussion: Page 24, line 3: Is the sentence “clinicians can reassure patients that CP after TKR is generally not permanent and that improvement is likely for many, if not most” true giving the findings? 12% did report CP and your data shows that patients fluctuate in and out of the CP category. I think the authors should reconsider if their data have enough validity to make such a statement	Thank you, we agree that it’s important to ensure that our statements are appropriate and align with findings. Given that the majority (85.3%) of individuals with CP at 1 year post primary report no-CP by year 5 we believe this is a fair statement although we agree that we would do well to highlight the fluctuation that many patients encounter over the years. To make that clearer, we have rephrased this sentence as follows: “In the meantime, clinicians can reassure patients that, though many fluctuate in their levels of pain, CP after TKR is generally not permanent and that improvement over the five year after their primary is likely for many, if not most” and added a similar note in the abstract.
Conclusion: Again, I would be cautious to state that most patients recover from TKR over time. It might be correct, but giving the limitations of massive drop-out, CP categorization based on a cut-off score from the OKS-PS, future research could very well report findings in contrast with the present results. I think the findings is more explorative than definitive and suggest reporting as such.	Thank you, we agree that caution is needed. To address this comment we have rephrased the conclusion and added text to highlight this.
Acknowledgements: A minor comment, but I find somewhat strange that several of the authors are also listed in the acknowledgement section? For me, the acknowledgement section is about acknowledging those that participating or contributed to the study, but not enough to qualify as authors.	Thank you for spotting this, we are sorry that this repetition occurred. The acknowledgements were a standard statement that had been produced for the COASt study. We have now removed author’s names from acknowledgement statement.
Reviewer 2	
1. Line 8, page 6: I suggest changing "osteoarthritis" to ""end-stage knee osteoarthritis"	Thank you, this is a helpful suggestion, and we agree that osteoarthritis that is at this stage may be seen as “end stage”. However, to ensure that our representation of osteoarthritis reflects diverse experiences and patients views we prefer and have added the term “advanced” instead. We hope that this is acceptable to the reviewer.
2. Lines 15-16, page 6: "There is currently limited understanding of how chronic pain (CP)" It is unclear to the reader whether or not CP is knee/post surgical specific or not so I suggest using "Knee OA associated CP, or post surgical CP?" This comment also applies to the rest of the paper.	We have now clarified in this sentence (in the very first paragraph of the introduction) that the study is referring to “chronic pain (CP) after a TKR”. This is followed by further references, in the next paragraph and elsewhere in the manuscript, to “CP in the first year after TKR” as a reminder.

3. It would be really useful and its important if you can add further patients characteristics such as presence or absence of deformities (fixed flexion and varus) both pre and post operatively as a surgical variable this is in addition to ACL (anterior cruciate ligament).	Thank you, we agree this could be useful from a contextual perspective, particularly for readers with background in surgery, unfortunately, we are unable to add this information because it was not available for this study.
--	---